

# Impacts of land-use management on ecosystem services and biodiversity: an agent-based modelling approach

Thomas J. Habib[1], Scott Heckbert[2], Jeffrey J. Wilson[3], Andrew J. K. Vandenbroeck[4], Jerome Cranston[1] and Daniel R. Farr[1]

[1] Alberta Biodiversity Monitoring Instutite, Edmonton, Alberta, Canada
[2] Alberta Innovates Technology Futures, Edmonton, Alberta, Canada
[3] Green Analytics, Guelph, Ontario, Canada
[4] Silvacom Ltd., Edmonton, Alberta, Canada

Corresponding author
Thomas J. Habib, thabib@ualberta.ca

## ABSTRACT

The science of ecosystem service (ES) mapping has become increasingly sophisticated over the past 20 years, and examples of successfully integrating ES into management decisions at national and sub-national scales have begun to emerge. However, increasing model sophistication and accuracy—and therefore complexity—may trade-off with ease of use and applicability to real-world decision-making contexts, so it is vital to incorporate the lessons learned from implementation efforts into new model development. Using successful implementation efforts for guidance, we developed an integrated ES modelling system to quantify several ecosystem services: forest timber production and carbon storage, water purification, pollination, and biodiversity. The system is designed to facilitate uptake of ES information into land-use decisions through three principal considerations: (1) using relatively straightforward models that can be readily deployed and interpreted without specialized expertise; (2) using an agent-based modelling framework to enable the incorporation of human decision-making directly within the model; and (3) integration among all ES models to simultaneously demonstrate the effects of a single land-use decision on multiple ES. We present an implementation of the model for a major watershed in Alberta, Canada, and highlight the system's capabilities to assess a suite of ES under future management decisions, including forestry activities under two alternative timber harvest strategies, and through a scenario modelling analysis exploring different intensities of hypothetical agricultural expansion. By using a modular approach, the modelling system can be readily expanded to evaluate additional ecosystem services or management questions of interest in order to guide land-use decisions to achieve socioeconomic and environmental objectives.

## INTRODUCTION

Ecosystem services (ES) are the benefits derived from natural systems that contribute to human well-being (*Millennium Ecosystem Assessment, 2005*). These benefits include tangible products such as food, fuel, and fibre, regulating services that make our
environment more liveable, and experiential values such as aesthetic appreciation and recreation. The practice of ES assessment has advanced rapidly in recent years, with methods of quantification progressing from early efforts to estimate the total value of ecosystem services across entire regions using estimates of value for different landcover types ("benefits transfer") (*Costanza et al., 1997*) to detailed models that capture the actual flow of services from ecosystems to people by explicitly linking ecological production functions to human users (*Kareiva et al., 2011*; *Bagstad et al., 2013a*). These mechanistic pathways range from straightforward, spatially uncoupled relationships such as the benefits of carbon sequestration for climate change mitigation, to the close spatial proximity required for insect pollinators to benefit crops, to directional flows such as wetlands that capture flood waters before they reach a developed area (*Costanza, 2008*).

Ecosystem services (ES) information is increasingly viewed as an important part of land-use planning and environmental management (*Daily et al., 2009*). The ability to simultaneously assess multiple ES and other related socioeconomic and environmental indicators such as food production and biodiversity, provides an understanding of the complex trade-offs associated with land-use (*Foley et al., 2005*). A variety of multi-ES assessment platforms exist or are currently under development, with varying levels of detail, quantification, generalizability, and usability (*Bagstad et al., 2013b*). However, if the goal of ES researchers is to influence decisions made by policy- and decision-makers, a model's complexity and functionality must be weighed against its ability to be used and understood by non-experts. The Natural Capital Project's InVEST toolkit (www.naturalcapitalproject.org) is one of the more widely-used modelling packages. In a review of their experience in applying InVEST to different land-use management issues in several jurisdictions, *Ruckelshaus et al. (2013)* described several attributes of modelling systems and processes that improved the uptake of ES information by decision-makers. Interestingly, their experience suggests that stakeholder ES goals are typically broad, such as ensuring an adequate supply of fresh water for agriculture, drinking, and biodiversity, rather than specific measurable objectives typically used in a formal optimization analysis. As such, the appropriate model outputs for these goals may be as simple as relative rankings that can identify priority management locations and activities. Therefore, relatively simple models that use a limited number of input parameters and can be run rapidly on standard desktop computers, but consequently sacrifice some amount of accuracy, precision, or spatial or temporal resolution, may be sufficient for this purpose. Simple models are also likely to be easily visualized and communicated, and therefore more amenable to the iterative science-policy processes most likely to yield on-the-ground results (*Ruckelshaus et al., 2013*).

Regardless of their complexity, a key capacity of ES models is the ability to estimate how the supply and value of ES change in response to management actions (*Ruckelshaus et al., 2013*). While representing management outcomes in a model can be accomplished by adjusting parameters to mimic altered management practices, a more holistic solution is to represent human decision-making behaviour as an integrated model component. Spatially-explicit agent-based models (hereafter "ABMs," typically termed "individual-based models" in ecology) are a useful tool for modelling ES, as they are well-suited to

represent the reciprocal interactions between ecological and socioeconomic systems that characterize ES (*An, 2012*; *Filatova et al., 2013*). The defining feature of ABM are "agents"; these are elements that exist in a model's landscape that have the ability to move, make decisions that can influence other model elements (cells or other agents) and processes, and contain and track variables. Typical examples of decision-making agents in social-ecological models include landowners, corporations, and individual animals. Incorporating the behaviour of these decision-making entities into a spatially explicit, cell-based model, adds considerable functionality for exploring the two-way linkages between socioeconomic and environmental systems (*Parker et al., 2003*; *Parker, Hessl & Davis, 2008*). For example, the potential supply of timber can be determined by the land cover, forest stand type and age, climate, and other growing conditions, all of which could be simulated in a purely cell-based model. However, converting the potential ES of standing timber volume to the final ecosystem service of wood products actually used by humans requires consideration of the actions of forestry companies that operate mills (i.e., agents). Additionally, the decision of where and how much timber to harvest will depend on environmental policies, wood product prices, and road networks, among other factors. Critically, the ABM structure allows for decisions to be represented at multiple geographic extents, including cell-level, regional (e.g., several forestry companies each control timber harvest in distinct areas), and global extents (e.g., an environmental policy affects forestry activities across the entire study area).

Integrated models that represent multiple ES and decision-making processes within a single, unified platform are capable of demonstrating how a variety of indicators respond to a single management action. Such models are essential to help decision-makers understand trade-offs among biodiversity, ES, and other socioeconomic indicators such as business revenues, and enable them to balance a variety of public and private values (*Nelson et al., 2009*; *Goldstein et al., 2012*). For example, decisions to harvest timber on public land will impact, at minimum, forestry company revenues, carbon storage, water flow and purification, recreational opportunities, and biodiversity.

While ABMs have previously been used to assess some aspects of ES as they relate to human land-use decisions (*Brady et al., 2012*; *Heckbert et al., 2012*; *Cong et al., 2014*; *Villamor et al., 2014*), most efforts have tended to include a limited number of ES, and may only include proxies for ES such as areas of landcover types rather than mechanistic models (i.e., production functions; *Kareiva et al., 2011*). However, detailed ABMs that simultaneously model multiple ES, human decision-making, and the mutual feedbacks between them can quickly become extremely complicated, and may be difficult to communicate within iterative science-policy processes. In consideration of this push-and-pull between ever-more detailed models and usability, we have developed a suite of integrated ecosystem service assessment models within an ABM platform, using relatively straightforward methods as recommended by the experience of *Ruckelshaus et al. (2013)*. We developed models to quantify forest timber production and carbon storage, water purification, pollination, and biodiversity. We provide a demonstration of the modelling system deployed in the North Saskatchewan River watershed in Alberta, Canada, as well as its capabilities to evaluate alternative management and land-use change scenarios.

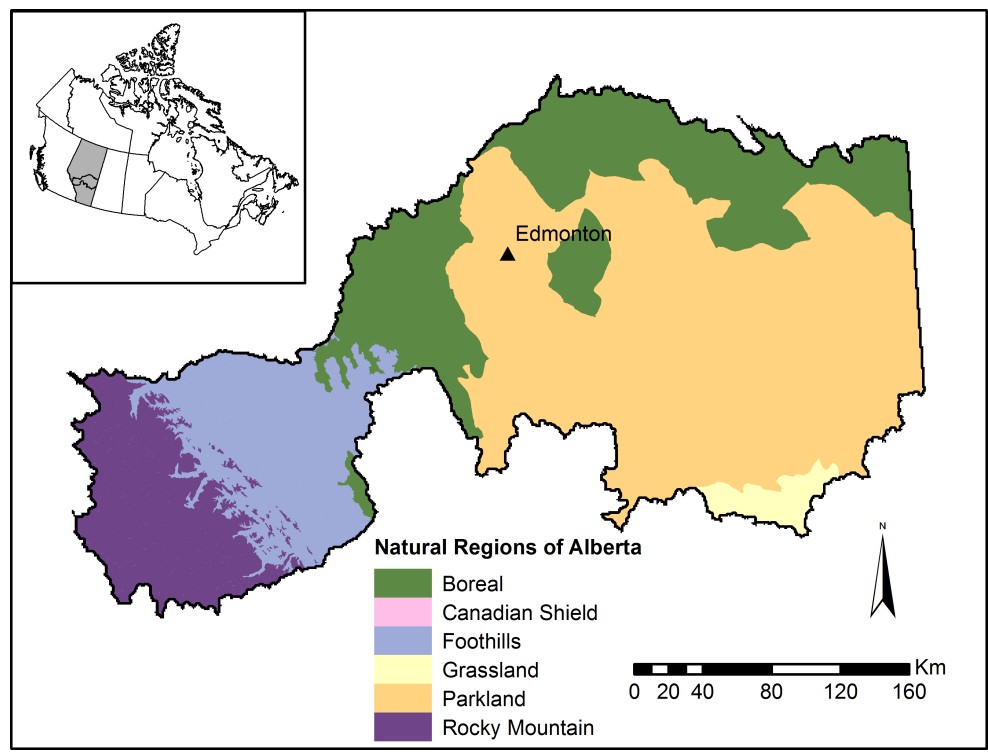

**Figure 1** **Study area in Alberta, Canada.** Region includes the North Saskatchewan River and Battle River watersheds in central Alberta. Inset map depicts the region's location within Alberta (grey area) and Canada.

## METHODS

### Study area

The North Saskatchewan River watershed is a major, 82,000 km$^2$ watershed in the province of Alberta, Canada, and is home to approximately 1.3 million people (*North Saskatchewan Watershed Alliance, 2012*), including the province's capital city of Edmonton (Fig. 1). The region extends across the entire province from west to east, and contains most of the major landcover categories of the province, including mountains, forested areas, parkland, agricultural regions, and urban centres (Fig. 2). Dominant land-use activities in the watershed include agriculture east of the mountains, forestry in the foothills, energy development, and urban development.

### Modelling platform and general model setup

We developed ES models using NetLogo, an open-source, freely available ABM platform (*Wilensky, 1999*). There are three principal elements in NetLogo: a uniform grid of cells that represents the landscape, user-defined types of agents, and links that can be used to form networks among agents. Both cells and agents can contain as many variables as needed. Models can contain any number of different types of agents. Under the Alberta Township System, parcels of land are divided into 1/2 -mile × 1/2 -mile "quarter-sections," and many land-use decisions are made at this scale; therefore, we used an 800 m (i.e., 1/2 mile) cell

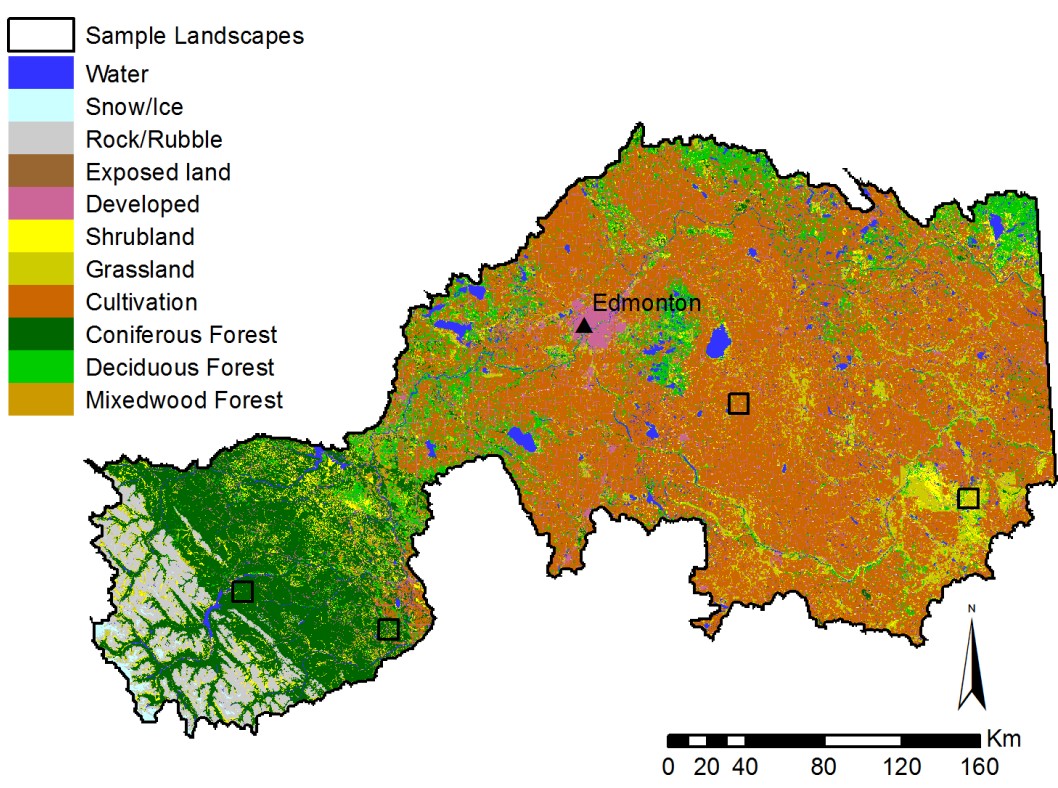

**Figure 2** **Major landcover types in the North Saskatchewan watershed region of Alberta, Canada.** Representative sample landscapes in unharvested forest, harvested forest, grassland, and agricultural areas are outlined in black.

size to represent this administrative system. While this is a relatively coarse scale for some processes (e.g., simulating surface water flow), using a coarser resolution is a necessary trade-off when modelling a study area this large; using a finer resolution would have been computationally prohibitive. However, the model code is flexible, allowing users to deploy it at any cell size that is appropriate for the study area extent and GIS data available, by changing the model variable "area" (see Table A in Supplemental Information 1).

In each grid cell, we calculated the area of each landcover type and human footprint type present within a cell. Thus, while the spatial location of each feature—and the spatial relationship between features—within cells is lost, the approach maintains information on features that are spatially small but may have a disproportionately large influence on the provision and value of ecosystem services (e.g., roads). Spatial landcover data (*Alberta Biodiversity Monitoring Institute, 2012a*; *Alberta Biodiversity Monitoring Institute, 2012b*; *Alberta Biodiversity Monitoring Institute, 2014a*) include both natural and anthropogenic features (see Table A in Supplemental Information 1). In addition to landcover data used by all models, cells also include relevant model-specific variables, as described below. All models operate at an annual time step. See File S2 for a NetLogo file containing the modelling system code.

## Forest timber and carbon

This model is based on a forest growth algorithm, with stands of timber altering their volume and carbon balance at each annual time step. We developed forest growth and carbon accumulation equations as functions of stand age by fitting polynomial curves to forest stand data from the Canada National Forest Inventory (*Natural Resources Canada, 2006*) and a published forest carbon model (Carbon Budget Model of the Canadian Forest Sector, known as CBM-CFS3; *Kurz et al., 2009*; *Kull et al., 2011*), respectively (see Supplemental Information 1). Based on these sources, different forest stand types exhibit different timber growth and carbon accumulation curves, and this is reflected in the model; forests are separated into deciduous, spruce-dominated or pine-dominated coniferous and mixedwood stands in each of two federally defined ecozones (Montane Cordillera and Boreal Plains) to align with the source data.

Forested land in Alberta is divided into administrative, non-overlapping Forest Management Units (FMU) that use multiple forms of tenure to allocate timber harvesting rights to companies or individuals (*Alberta Agriculture and Forestry, 2016*). Large-scale, long-term, area-based Forest Management Agreements (FMA) are the most significant form of tenure, representing 63% of the timber harvested in Alberta (*Alberta Agriculture and Forestry, 2014a*). Volume-based coniferous and deciduous Timber Quotas are another significant form of tenure for the forest industry, representing 25% of timber harvest, much of it within FMUs governed by FMAs. The remaining timber volume is harvested through Timber Permits, industrial salvage, and non-commercial activities (*Alberta Agriculture and Forestry, 2014a*). The present model focuses on FMA-based timber harvest, as well as Timber Quotas within FMA areas; combined these two forms of tenure represent approximately 85% of timber harvest in Alberta.

In this model, timber in each FMU is directed to the corresponding mill (pulp or sawmill) operated by the company with tenure rights in that area. For each FMU, a sustainable amount of timber available for harvest in a year, known as the Annual Allowable Cut (AAC), is set out in publicly available forest management planning documents. Working at an annual time-step, the model simulates the spatial pattern and scheduling of timber harvest over several years, with the goal of maximizing profits while harvesting forest stands of the appropriate age (in this case, $\geq 80$ years old). Harvest decisions are made by agents representing mills, who harvest timber from eligible forest stands (i.e., cells) in the appropriate FMUs until the AAC for the year is reached; in reality, the full AAC is typically not harvested due to economic or operational reasons, and the model accounts for this by only harvesting a percentage of the AAC that is drawn from a normal distribution based on historical harvest rates (*Alberta Agriculture and Forestry, 2014b*). When a cell is selected for harvest, all of the timber within it is removed, representing the traditional clearcutting technique that dominates forestry in Alberta (*Alberta Agriculture and Forestry, 2014c*).

In the model, in order to maximize profits, mills harvest preferentially from stands where the cost to transport timber to the mill is the lowest. Transportation cost was based on the least-cost pathway calculated in ArcGIS (Spatial Analyst; Distance toolset). Because the primary factor driving transportation cost is travel time required due to fixed hourly costs including worker wages, the cost surface accounts for transportation speed by

considering whether each cell contains high-speed paved roads, moderate-speed unpaved roads, or other low-speed corridors that can be used to transport timber, such as pipelines or cut-lines (Table C in Supplemental Information 1). Once the AAC is reached in each year, mills process the harvested raw timber into wood products based on the mill type (i.e., pulp and paper, lumber, or oriented strand board), and monetary value is calculated based on the market price of each wood product and the operational costs of harvest and processing.

Carbon stored in forests is estimated annually, and is affected in the model by timber harvest activities. When timber is harvested from a cell, its age is reset to zero, and the associated amount of carbon, both above- and below-ground, is lost from the environment. However, the model does not address the rate at which carbon sequestered in wood products is released back to the atmosphere as $CO_2$, which can vary widely, from years to decades, depending on the wood products and their use (*Skog, 2008*; *McKinley et al., 2011*).

Additionally, this model does not incorporate the influence of natural disturbances such as wildfire. Fire is the dominant process shaping the boreal forest, and can play an enormous role in both forest harvest planning and carbon flux. However, despite the existence of sophisticated fire prediction (*Parisien et al., 2005*) and spread models (*Tymstra et al., 2010*), the location and size of future fires remains inherently unpredictable. Further, while incorporating a stochastic fire modelling process would be necessary to predict hypothetical future landscapes, the primary objective of this ES modelling suite is to understand current ES values and how they will respond to human management actions, rather than simulating future landcover scenarios.

We ran the simulation for a period of 20 years. Model outputs include the estimated net present value (NPV) of timber harvesting over this time period, the amount of carbon stored above- and below-ground in forests at the end of the 20-yr simulation, and the change in carbon storage over the simulation.

## Water purification model

The water purification model simulates precipitation, overland flow, and surface water flow over a single year, based on the landcover and climate at that point in time. In particular, the model was designed to identify areas contributing to non-point source export of nutrients and sediment (i.e., through surface runoff and erosion), important areas for removing these substances, and impacts to downstream water users. The current model implementation focuses on nutrients (nitrogen, phosphorus, and total suspended solids—TSS) and eroded sediment, which are major determinants of water quality. To set up the model, we used the Spatial Analyst (Hydrology) toolset in ArcGIS 10.1 (ESRI, Redlands, CA) to create a river network based on a 100-m digital elevation model with all sinks filled. After importing the river layer into NetLogo, we created links between all cells containing rivers to form the river network, allowing the model to represent the downstream movement of water and associated nutrient and sediment. We also created agents at points of interest such as tributary outlets and municipal water treatment plants to track model output variables, including water flow and the cumulative annual load of each water quality indicator.

We obtained mean annual precipitation for each cell, based on 1971–2001 climate normals from ClimateWNA (*Wang et al., 2012*), and calculated runoff from each cell as a percentage of total precipitation, based on runoff coefficients for each landcover type (*Donahue, 2013*). Because each cell contains multiple landcover types, each cell's runoff coefficient is calculated as the area-weighted sum of runoff coefficients for each landcover type. After the precipitation event, the volume of water that runs off each cell, represented as a "raindrop" agent, moves to the adjacent cell with the lowest elevation, and this downslope movement continues until it reaches a cell that intersects the river network. At this point water flow moves downstream along the connected river network.

Non-point source export of nutrient and sediment are loaded into surface water flow through two different modelling processes. We used export coefficients (measured in kg ha$^{-1}$ mm$^{-1}$ of annual precipitation) calculated for major landcover and human footprint types (*Donahue, 2013*) to estimate the amount of N, P, and TSS loaded into surface runoff (hereafter, "loading"). Similar to calculating runoff, we used the area-weighted sum of export coefficients for all landcover types present in a cell to calculate the total amount of each nutrient released. Sediment erosion was estimated using the Revised Universal Soil Loss Equation (RUSLE), which estimates erosion based on rainfall, soil characteristics, slope, and land management (*Renard et al., 1997*). To select the parameters for RUSLE, we relied on published guidance for Canada (*Wall et al., 2002*) and obtained the necessary soil data from Agriculture and Agri-Food Canada's Soil Landscapes of Canada version 3.2 (*Soil Landscapes of Canada Working Group, 2010*) for our study region.

Nutrient removal occurs during overland flow, where a percentage of each raindrop agent's nutrient and sediment load is removed as it flows across downslope cells before reaching the river network. Nutrient and sediment removal percentages are assigned for each landcover type, and the total amount removed by a cell is the area-weighted sum of the landcover types comprising the cell. In addition to producing output maps for nutrient and sediment loading and removal, the model also calculates the amount of nutrients and sediment actually supplied to the river network by each cell; that is, the amount a substance contained in a cell's runoff that eventually makes its way into the river network (i.e., is not retained at some point between the origin cell and the river network). Combined, these three processes predict areas that have important effects on water quality, as well as identifying priority management areas. In addition to the output maps, the model also outputs a table of water variables that includes the cumulative annual flow and load of nutrients and sediment at water monitoring points of interest along the river network.

We calibrated the water purification model using cumulative annual phosphorus load data from 8 tributary basins in the region; the observed water data were estimated from periodic water samples taken from 1985–2008, with the number of observations at each monitoring point ranging from 11 to 241 (mean 83). We conducted a global calibration, varying each of the nutrient export coefficients with 10 unique parameter sets obtained using the Latin Hypercube sampling method (*McKay, Beckman & Conover, 1979*; *Abbaspour et al., 2007*). Distinct parameter sets were selected for each of three Natural Region areas (Mountains, Foothills, and the same parameters for the Boreal, Parkland, and Grassland, all of which are topographically and geologically similar as part of the Western Canadian

Sedimentary Basin, and largely contain similar landcover within the study area). Please refer to the Article S1 for more information on the calibration procedure and results.

## Pollination

The pollination model estimates the additional yield of canola that can be attributed to native insect pollinators. Although several local crops may benefit from insect pollination, canola is by far the most valuable Alberta crop that receives such benefits; in 2014, farm cash receipts for canola were $2.54B, representing 43% of Alberta's total crop revenues for that year (*Statistics Canada, 2015a*). The next-most valuable crop, wheat ($2.1B, representing 29% of Alberta's 2014 crop revenues), does not benefit from insect pollination. Thus, most direct economic benefits of pollination on Alberta crop production are likely realized through canola production, based on the dominance of canola as a cash crop in Alberta.

Annual crops such as canola can provide a large food resource for insect pollinators, but only during a short period of time while in bloom. Additionally, annual crop fields that are disturbed each year do not provide nesting habitat for ground-nesting bees, which are the most common native pollinators in Alberta (*Sheffield, Frier & Dumesh, 2014*). In order for a crop field to receive benefits from wild pollinators, it must be within the foraging distance of pollinators occupying nesting habitat, such as undeveloped areas and perennial tame pasture. Pollinator visitation rates and stability of pollination services have been linked in many areas to proximity to natural or semi-natural areas across multiple crop systems (*Ricketts et al., 2008*; *Garibaldi et al., 2011*). In Alberta, increases in canola yield have been linked to the abundance of uncultivated land within bee foraging distance (generally <1 km), which provides bee nesting habitat (*Morandin & Winston, 2005*; *Morandin et al., 2007*).

We used data from *Morandin & Winston (2006)* to develop a field-level model of canola yield as a function of bee abundance; in turn, bee abundance was estimated as a function of the amount of natural (i.e., undeveloped) and semi-natural (i.e., tame pasture) land within neighbouring cells, representing the amount of bee nesting habitat within foraging distance of each canola field (see Article S1 for equations). This aligns well with *Morandin & Winston (2006)* who defined pollinator nesting habitat within a 750-m buffer around each field. We delineated the boundaries of crop fields using vector-based landcover data (*Alberta Biodiversity Monitoring Institute, 2012a*; *Alberta Biodiversity Monitoring Institute, 2012b*), as well as small uncultivated areas within them which may represent important pollinator nesting habitat. To represent agricultural polygons on which canola was grown, we used annual crop maps (2009–2012; resolution 30 m or 56 m) created by *Agriculture and Agri-Food Canada (2012)*. The model runs at an annual time step; because different crops are typically planted in a given field each year, we used 4 years of crop data to represent typical canola rotations (*Alberta Agriculture and Forestry, 2012*). Therefore, as the model runs for multiple years, the locations of canola fields change at each time step, repeating every 4 years. In each year of the simulation, the model estimates the additional crop yield due to pollinators, and the monetary value is calculated by multiplying this by the user-defined canola price. When run for multiple years, the economic value is expressed as

the net present value, based on a user-defined discount rate; we used a rate of 2% for the results presented here.

## Biodiversity index

To assess biodiversity, we used field observations obtained by the Alberta Biodiversity Monitoring Institute to create a biodiversity index to express the degree to which ecological communities are impacted by anthropogenic disturbance (*Nielsen et al., 2007*). The original data (*Alberta Biodiversity Monitoring Institute, 2015*) were used to develop detailed species abundance models as a function of landscape characteristics (*Alberta Biodiversity Monitoring Institute, 2014b*). For each species, the abundance model was run for two landscapes: the current landscape, and a hypothetical "reference" landscape with all human footprint removed and backfilled with the landcover types that used to be present (*Alberta Biodiversity Monitoring Institute, 2014a*). The difference between current and reference abundances for each species is then expressed as a percentage, where a 100% score represents no change from its reference abundance, and the score decreases as the current abundance deviates from the reference condition. Both declines and increases (i.e., overabundance) lead to lower index values (*Alberta Biodiversity Monitoring Institute, 2014b*). We averaged the index scores for 447 available species (comprising 80 birds, 13 mammals, 202 vascular plants, 90 bryophytes, and 62 soil mites) to obtain a single biodiversity index value, which is a representation of how much the entire community has changed due to human activities. Because the overall biodiversity index is an average, individual species responses will be more variable than the overall index suggests; therefore it cannot be used to support decision-making related to individual species of particular management concern, such as species at risk. Instead, it should be used as a coarse filter for biodiversity-related objectives, and complemented by analyses or models of priority species as required for a given management decision. There is no temporal element in the biodiversity model; rather, it estimates the biodiversity index for a single point in time based on landcover characteristics.

To deploy the biodiversity model in the suite of ES models, we regressed overall biodiversity index for each cell against the amount of different types of human footprint present (categories included roads, trails, forestry cutblocks, agriculture, and industrial developments) to obtain an overall footprint-biodiversity index equation. This simplified equation (see Supplemental Information 1) fits the full model very well ($r^2 = 0.94$), and allows the ES modelling package to rapidly assess biodiversity condition. The model output is the biodiversity index score for each cell of the study region, measured from 0 to 100%.

## Model integration and land-use management decisions

Each model differs in terms of what modelling elements (i.e., cells, agents, and links) are used; the water purification model draws on all three, the forest timber & carbon model uses mill agents to control timber harvest, and the pollination and biodiversity models are purely cellular models. Despite the varying amounts of human behaviour directly incorporated into each model, the integrated nature of the modelling system allows users to evaluate how a given management decision may influence multiple ES; we demonstrate these capabilities of integrated ES assessment models in three ways.

First, we ran the water purification and biodiversity models both before and after simulated timber harvest to understand how non-point source nutrient runoff and biodiversity are affected by timber harvest activities. Additionally, we drew on the ABM properties of this model to allow mills to conduct timber harvest activities according to either traditional clearcutting, or variable retention harvesting (*Serrouya & D'Eon, 2004*), which may have additional benefits for biodiversity (*Fedrowitz et al., 2014*). We also demonstrate how this change in forestry practices may impact the NPV of timber harvest. We conducted two simulations, one with all mills using clearcutting such that any harvested cell had all of its timber removed, and one with all mills using variable retention techniques, in which a cell selected for harvest had either 25%, 50%, or 75% of its timber volume harvested, with the percentage chosen randomly; this range of harvest intensity is typical of large-scale management experiments (*Serrouya & D'Eon, 2004*). While variable retention techniques can include altering both the intensity and spatial pattern of harvest, we only modelled changes in intensity, because fine-scale patterns (e.g., 10 ha, 1 ha, and 0.1 ha patch cut arrays; (*Huggard & Vyse, 2002*) cannot be represented in an 800m cellular landscape. We omitted pollination from this analysis, as crop production does not occur in the same areas as commercial timber harvest.

Second, we highlighted the suite of ES provided in four typical 100 km$^2$ landscapes representative of major land-uses in the study region: unharvested forest, nearby areas that have been harvested for timber, native grasslands, and an agricultural area of annual cropland (Fig. 2). To compare pollination, timber, water purification, and carbon storage across landscapes, we standardized each of these ES by expressing each landscape's value as a percentage of the maximum value estimated across all four landscapes; for the water model, we chose to compare only one output variable (the amount of phosphorus supplied to the river network). The biodiversity index is already calculated as a percentage, so no additional calculations were necessary to compare across the four landscapes. This analysis can be considered to show how historical land-use decisions have shaped ES provision across the region.

Finally, we conducted a scenario modelling analysis to evaluate the impact of potential future agricultural expansion on multiple ecosystem services. The dominant historical land-use change in the study area has been agricultural conversion to produce crops (Fig. 2). While much of this landscape change occurred over the 20th century, a significant amount of conversion has continued more recently; from 2000–2012, nearly 3 million hectares were converted to cropland province-wide, most of which (2.4M ha) was formerly pasture (*Haarsma, 2014*). We simulated future agricultural expansion by iteratively converting a proportion of the region's remaining pasture into cropland, in 5% increments. In each simulation, we set the conversion proportion and compared it to a randomly drawn number between 0–1 for each cell; if the random number was lower than the pre-determined conversion proportion, then the cell's pasture was converted. We selected pastures for conversion randomly, which is less realistic than using a probabilistic model based on environmental and socioeconomic factors linked to higher conversion rates. However, the vast majority (77%) of pastures in the study area occur on soils suitable for crop production

(Class 2 and 3 under Alberta's Land Suitability Rating System; *Agriculture and Agri-Food Canada, 1995*), which *Ruan, Qiu & Dyck (2016)* found to be the most important factor leading to the conversion of pastures to cropland in Alberta. All newly converted cropland was randomly assigned to grow canola in one year out of a 4-year rotation. For each level of simulated landcover change, we ran the ES models to estimate the total regional amounts of pollination value, total canola revenue, phosphorus runoff, and biodiversity. Forested areas subject to commercial timber harvest do not overlap with this type of landscape change, so we omitted the forest timber and carbon model from this analysis.

## RESULTS

### Current values of ecosystem services

#### Forest timber & carbon model

Over the 20-year simulation under standard clearcutting practices, the estimated NPV of timber within the study area was projected to be $719M (Fig. 3A). While forest sector statistics are not available for specific watershed regions, a 1-year simulation for the year 2010 across the entire province estimated timber revenues of $3.32B (Table B in Supplemental Information 1), representing 91% of the recorded total provincial revenues for wood and pulp products of $3.65B for that year (*Statistics Canada, 2015b*); given that the model only captures the 85% of timber harvested within Alberta (i.e., FMA-based harvest, including Timber Quotas within FMA areas), the model output aligns reasonably well with the observed forestry revenue from that year.

Forest carbon storage under current conditions was estimated at 2.5B tonnes of $CO_2$-equivalent (Fig. 4). Over the 20-year period, an estimated 170M additional tonnes of carbon were sequestered in forested areas, even after accounting for the removal of carbon due to timber harvest. This represents an increase in carbon storage of 6.7% over the 20-year period (Fig. 5).

#### Water purification model

Non point-source export of nutrients and sediment varied dramatically across the study region, highlighting areas that are more (or less) important for determining water quality, and where management could be prioritized (e.g., Fig. 6; see additional Figs. E–J in Supplemental Information 1). However, to estimate the benefits of water purification actually received by people, it is necessary to identify not just source areas for excess nutrients and sediment, but also what areas impede this harmful flow prior to reaching a service area (*Bagstad et al., 2013a*). Therefore, we highlight the amount of each substance retained on the landscape contributing to the water supply of the City of Edmonton (Fig. 7), which represents the majority of the region's demand for clean drinking water. In an average year, the model predicts that the contribution area upstream of Edmonton retains 4.4M kg of nitrogen, 682K kg of phosphorus, and 466M kg of total suspended solids.

We calibrated the water model using the modelled cumulative annual nutrient loads at points in the river network and observed data for the same locations; only 8 locations in the region had sufficiently frequent water quality data to calculate the annual nutrient loads.

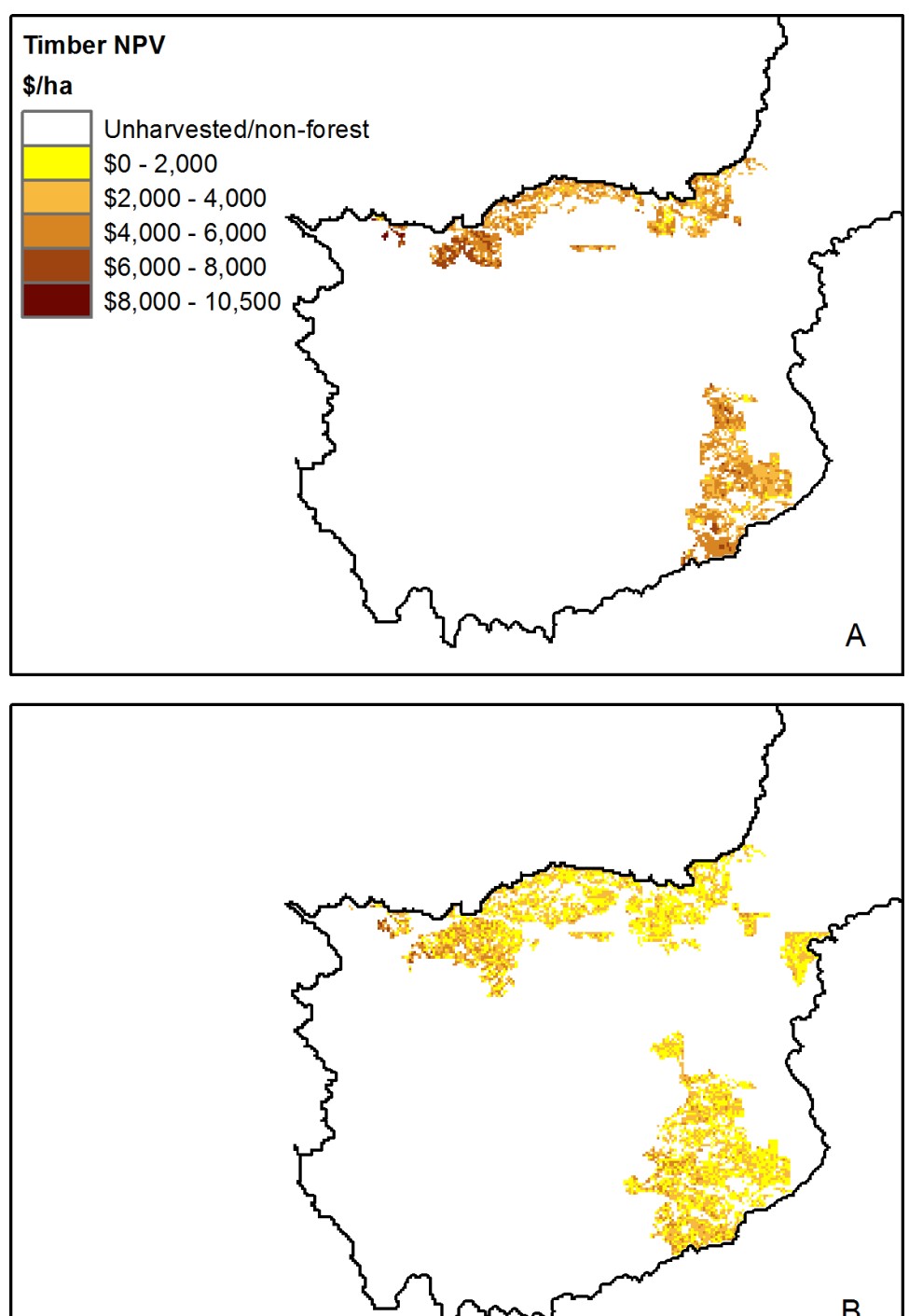

**Figure 3  Modelled net present value of 20-year simulated timber harvest.** Modelled timber net present value in the active forestry area of the North Saskatchewan watershed region of Alberta, Canada, based on a 20-year forest harvest simulation under (A) standard clearcutting practices, and (B) variable retention timber harvest. Map is restricted to the active forestry area of the North Saskatchewan watershed.

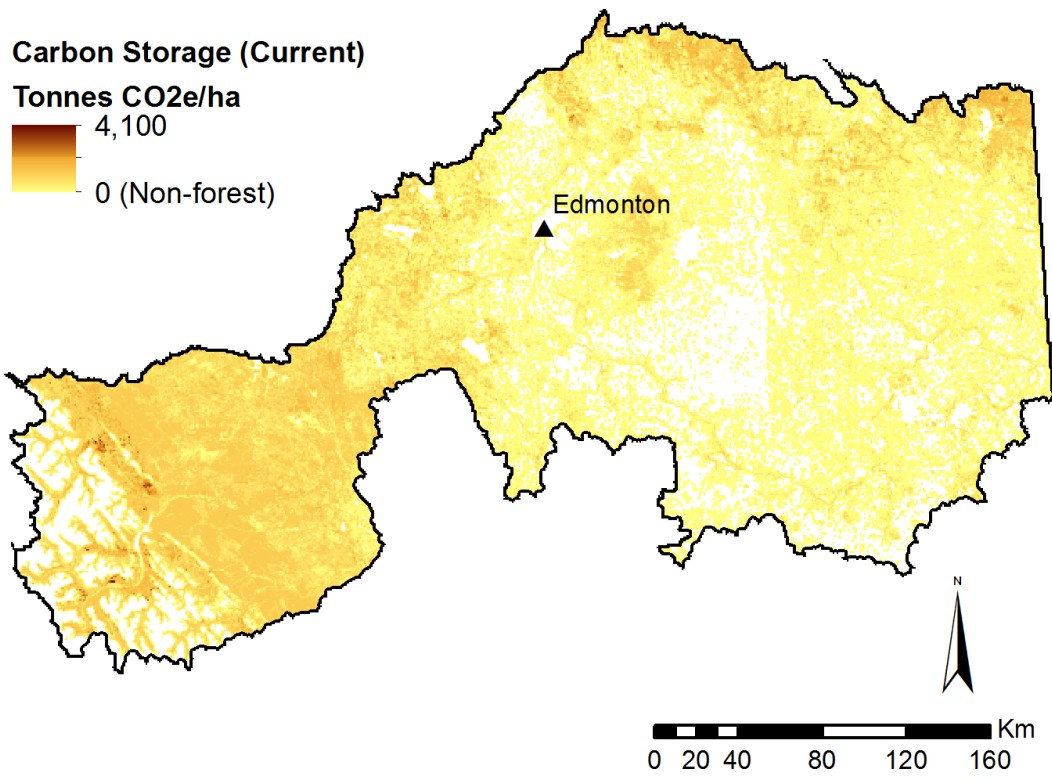

**Figure 4** **Modelled forest carbon storage under current (2010) landscape conditions.**

The model performed better in agricultural areas compared to the western mountainous region, which is likely due to the origin of nutrient export coefficients we used, which were calculated predominantly from low-order streams in agricultural areas (*Donahue, 2013*).

### Pollination model

Based on a four-year simulation and the 2010 crop price $461.81/tonne (*Canola Council of Canada, 2015*), we estimated the value of wild pollinators to canola production in the study area to be $971M, or up to $67,000 per quarter section over that time period (Fig. 8). Farm cash receipt data are only available at the provincial level, but based on similar scenarios run for the entire province, the amount of income attributable to pollinators ($2.65B; Table E in Supplemental Information 1) represents approximately 30% of revenue from canola production over the 2009–12 time period (*Statistics Canada, 2015a*).

The average canola yield predicted by the model was 27.3 bushels/acre, which is within the range of yields observed in Alberta from 2002–2003 (*Graf, 2013*) when the original data were collected by *Morandin & Winston (2006)*. However, Alberta canola yields have increased by 3.4% annually since 2000 (*Graf, 2013*), and therefore this estimate is low compared to observed yields in more recent years (e.g., 30.8 and 37 bushels/acre in 2009 and 2010, respectively; *Alberta Agriculture and Forestry, 2010*; *Alberta Agriculture and Forestry, 2011*). Collection of new field data may help account for additional explanatory factors, including interactions between native and managed pollinators such as honeybees, variability among different strains of canola, differences in pollinator communities,

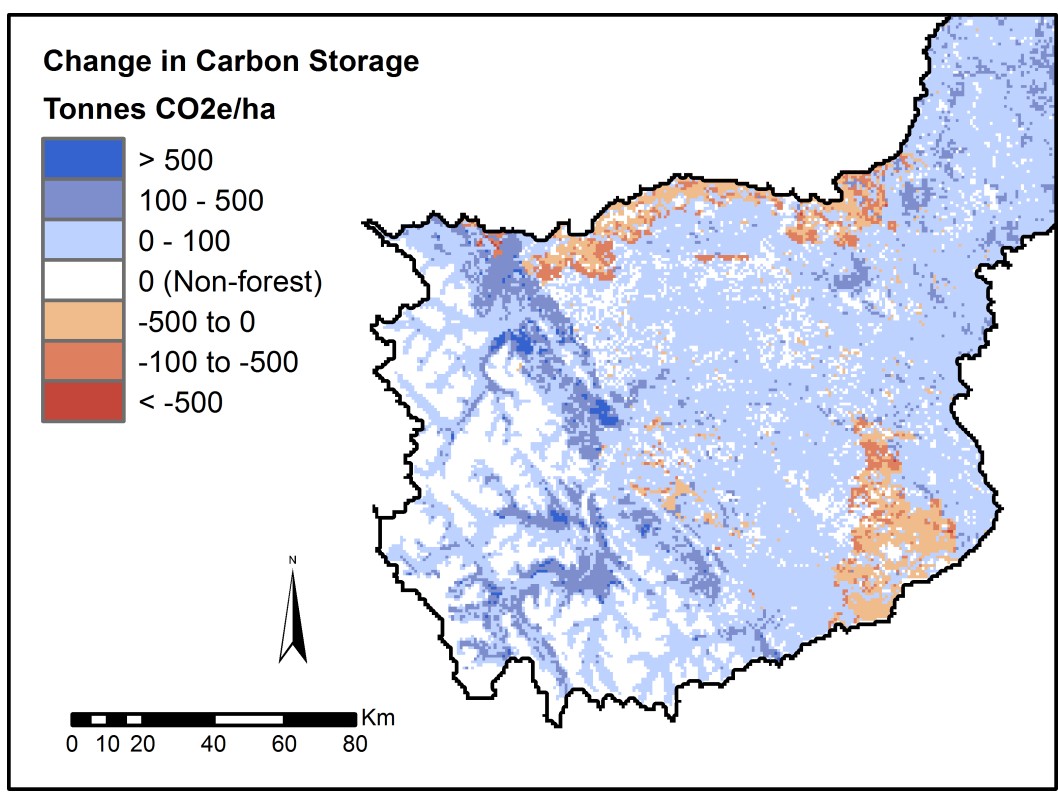

**Figure 5  Modelled change in forest carbon storage following timber harvest.** We simulated 20 years of timber harvest under standard clearcutting practices, and recorded the change in carbon storage from a baseline of 2010. Map is restricted to the active forestry area of the North Saskatchewan watershed.

climate, and landcover conditions across Alberta, and potential negative effects of planting canola in continuous or very short rotations (*Cathcart et al., 2006*; *Harker et al., 2014*). Additionally, conducting manipulative experiments comparing canola yield inside and outside of pollinator exclosures would isolate the contribution of pollination and provide a dataset to validate this or any other pollination model.

### *Biodiversity model*

Under current conditions, the average biodiversity index for the entire study area was estimated at 51.6%. The index score varied regionally (Fig. 9), from the nearly completely undeveloped national parks in the western mountains (99% biodiversity score), to the adjacent foothills region (77% biodiversity score), to urban areas and widespread agriculture in the central and eastern portions of the region (40% biodiversity score).

### Effects of timber harvest

The timber NPV under variable retention was estimated at $693M (Fig. 3B), a decrease of $26.7 million compared to the clearcutting simulation, representing 3.7% in foregone profits over the 20 year time horizon (Fig. 10). The 20-year clearcutting simulation led to an approximately 1% increase in nitrogen and phosphorus supplied to the North

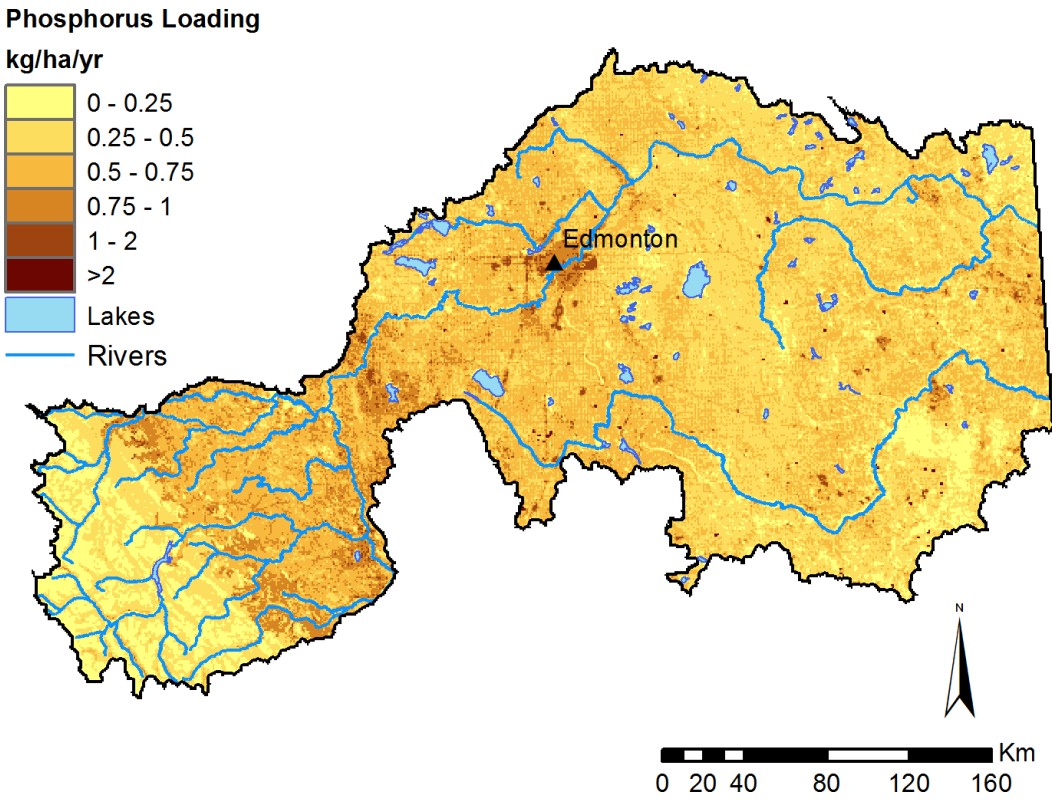

**Phosphorus Loading**
**kg/ha/yr**

| | |
|---|---|
| | 0 - 0.25 |
| | 0.25 - 0.5 |
| | 0.5 - 0.75 |
| | 0.75 - 1 |
| | 1 - 2 |
| | >2 |
| | Lakes |
| — | Rivers |

**Figure 6  Modelled phosphorus loading into surface runoff.** Phosphorus supply represents all phosphorus released by the landscape. Loading is based on average precipitation from the 1971–2001 climate normals. Major lakes (>300 ha) and rivers are also depicted.

Saskatchewan River, and a 3% increase in TSS supply; the effects of variable retention harvest on water and carbon model outputs were similar to those of clearcutting (Fig. 10).

Following 20 years of simulated timber harvest, the biodiversity index decreased in areas impacted by harvest activities. Unsurprisingly, moving from clearcutting to variable retention timber harvest leads to biodiversity impacts that are more extensive but less locally severe (Fig. 11). However, while the model is capable of predicting the spatial pattern of biodiversity responses to variable retention timber harvest, it is inappropriate for assessing the actual index values in areas subject to this type of timber harvest. This is because the mosaic of harvested and standing trees created by variable retention logging essentially represents a distinct new landcover type, rather than simply a smaller area of clearcut as the model currently treats it. Because this type of disturbance has less severe impacts on many species (*Schieck & Song, 2006*) the biodiversity index scores will be artificially low.

## Variation in suites of ecosystem services

The differences in the suite of ES provided by the four representative landscapes (grassland, agriculture, unharvested forest, and harvested forest) are dramatic, and all four landscapes show a marked specialization toward a subset of ES (Fig. 12). Note that a high "phosphorus runoff" score indicates more runoff, and therefore lower provision of clean water. All five axes in Fig. 12 are scaled from 0–100% for ease of comparison, rather than showing

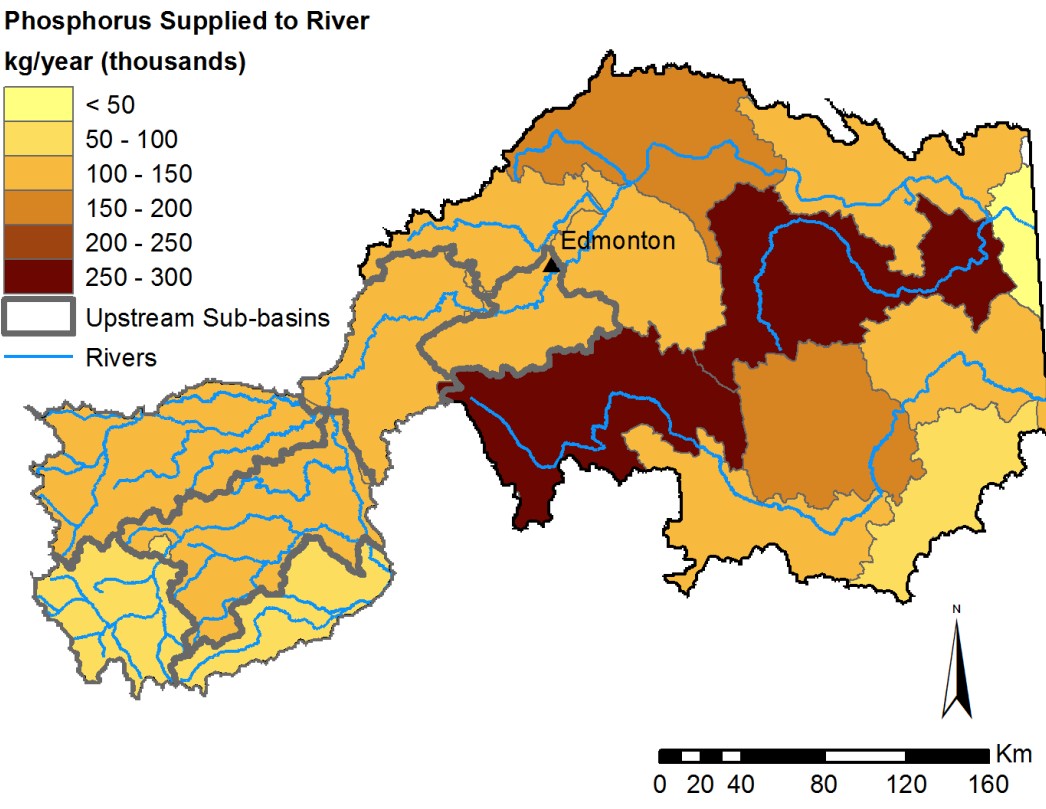

**Figure 7  Modelled phosphorus supplied by sub-basins to the river network.** Phosphorus supply represents all phosphorus released by the landscape that is not subsequently retained by the landscape during overland flow. Phosphorus supply is based on average precipitation from the 1971–2001 climate normals. Sub-basins upstream of the City of Edmonton are outlined in grey.

absolute ES values; therefore, direct comparisons between different ES are inappropriate, but the changes in a given ES across regions demonstrate the trade-offs associated with each land-use. For example, comparing the logged and un-logged forested landscapes, conducting forestry activities to obtain timber value was associated with relatively small decreases in carbon storage and biodiversity, but a marked increase in phosphorus runoff. Similarly, the historical shift in the eastern part of our study area from native grasslands to annual cropland shows a dramatic increase in phosphorus runoff, and a considerable decrease in the biodiversity index (Fig. 12).

## Agricultural expansion scenario modelling

In general, converting pasture to cropland for canola production increased total crop revenues and total pollinator value, but with diminishing returns as the proportion of land converted increased, and even declining in the case of pollinator value (Fig. 13). At all levels of conversion, the mean pollinator value per field decreased as pasture—and therefore pollinator nesting habitat—was removed (Fig. 13). Water purification declined across the landscape, with an increasing amount of nutrients loaded into runoff, and decreased capacity to filter this runoff (Fig. 14). Finally, the biodiversity index declined approximately linearly (Fig. 14).
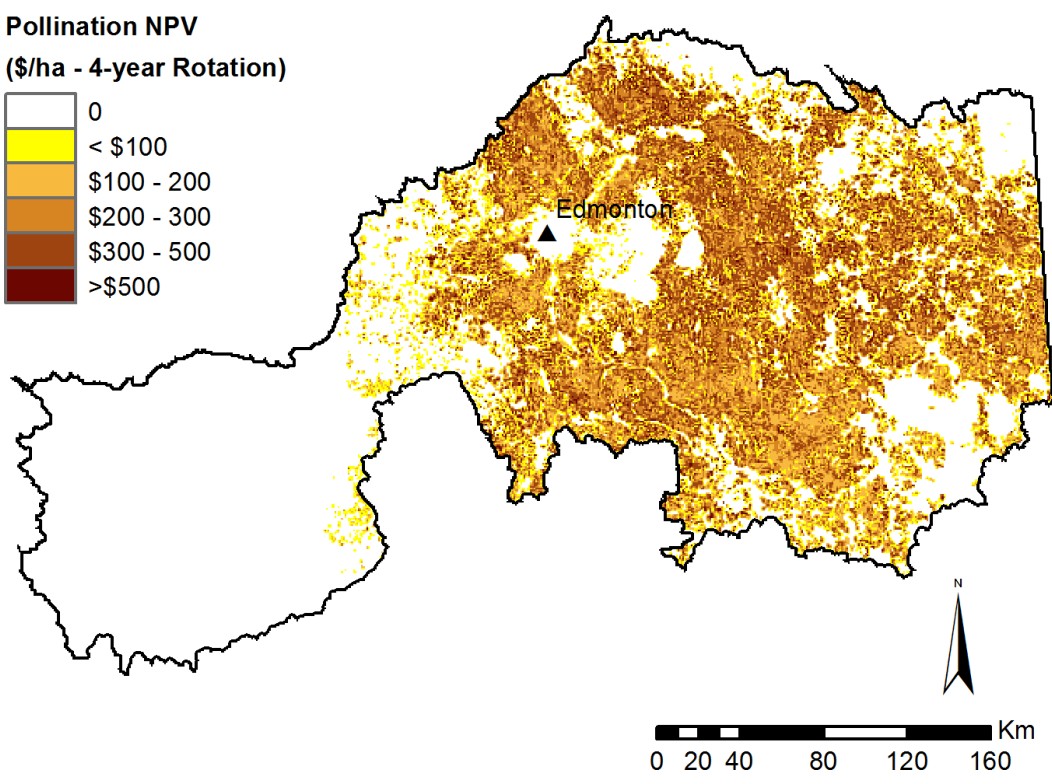

**Figure 8** **Modelled net present value of pollination for canola production.** Modelled pollination value is based on a 4-year crop rotation using crop type maps from 2009–2012.

In addition to the general trends, we highlight the results at the 40% conversion level, corresponding approximately to the observed conversion rate from 2000–2012 within our study area (*Haarsma, 2014*). This scenario corresponded to approximately 400,000 hectares of pasture being converted to cropland, representing a 15% increase in annual cropland area. In this conversion scenario, we found a 9% increase in total canola revenue, a 6% increase in total pollination value, but a 1% decline in average pollinator value per cell; that is, the lower number of pollinators per field was offset by the increase in area under canola cultivation. Within the portion of the study area dominated by agriculture (comprising the parkland, grassland, and boreal natural regions in Fig. 1), total phosphorus supplied to the river network increased by 3.1%, and the biodiversity index declined by 1.5 points, from 39.9% to 38.4%.

## DISCUSSION

### Integrated ecosystem services models and trade-offs

By combining multiple ES models within an ABM platform, we were able to quantify the trade-offs that are inherent to any land-use management activity, but typically hidden (Figs. 10 and 12). When new management practices are being considered, understanding these trade-offs is a critical part of the cost-benefit analysis. For example, shifting from clearcutting to variable retention timber harvest results in foregone profit for forestry

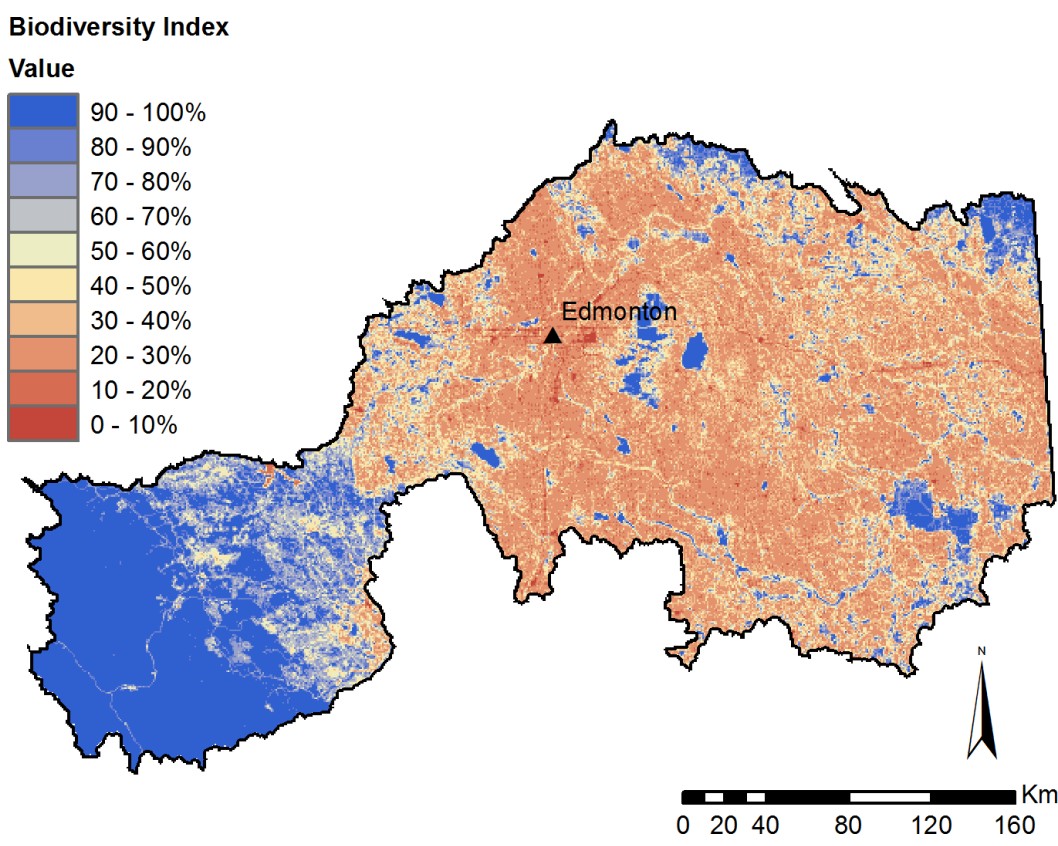

**Biodiversity Index**
**Value**
- 90 - 100%
- 80 - 90%
- 70 - 80%
- 60 - 70%
- 50 - 60%
- 40 - 50%
- 30 - 40%
- 20 - 30%
- 10 - 20%
- 0 - 10%

Edmonton

**Figure 9** **Modelled biodiversity index under current (2010) landcover conditions.**

companies; the integrated model results provide valuable information to help decision-makers determine whether the gain in other ES justifies this monetary cost.

Trade-offs may include changes in the production of ES, but also the abilities of people to access and use existing ES. In the provided example of agricultural expansion, adding new crop fields reduces the biophysical supply of an ES (loss of pollinator habitat), but—to a point—increases the ability of people to benefit from the service by creating more crop fields to receive those pollination benefits. In this case, there is a one-to-one exchange between land providing service provision and receiving benefits, as every parcel of new crop acreage comes at the expense of an equal area of pollinator nesting habitat. While the relationship between increasing access and the biophysical supply of a given ES will not always be in direct competition—or even related at all (*Wieland et al., 2016*)—this example demonstrates the complexities of evaluating the wide-ranging effects of land-use management, and underscores the need for models to consider both the biophysical supply of ES, as well as how they are actually used by people.

When considering implementing a land-use management action, it is important to understand how intensely a given management action should be applied in order to achieve the desired environmental and/or socioeconomic objectives. By providing the ability to quickly run multiple iterations of a scenario at different conversion rates, this modelling

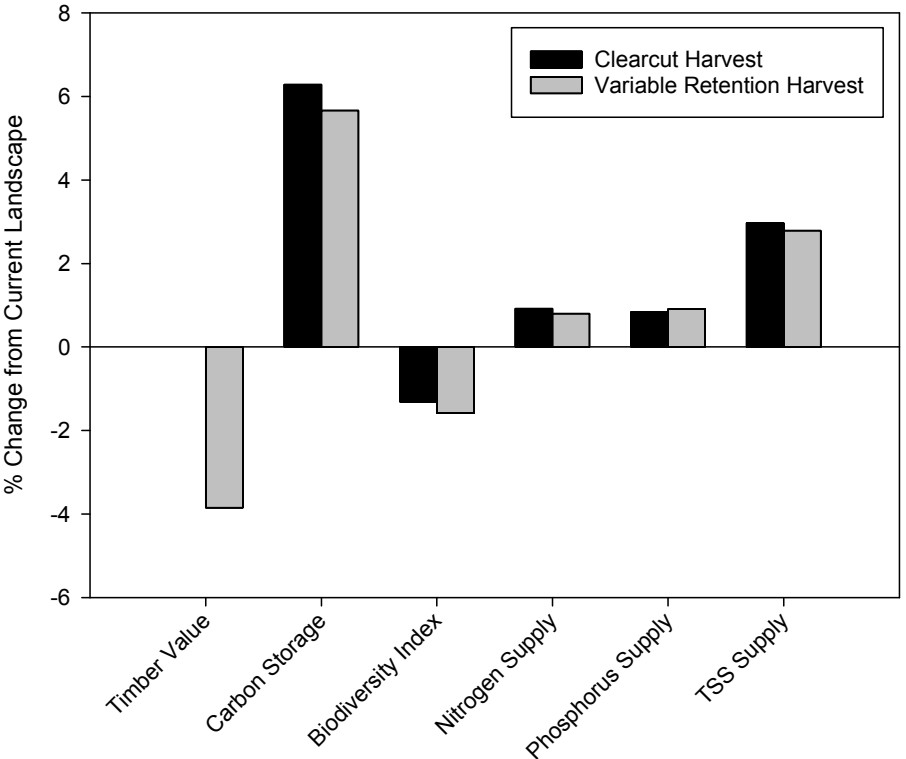

**Figure 10  Comparison of clearcutting versus variable retention timber harvest.** Modelled impacts of two timber harvest strategies on a suite of ecosystem services compared to current landscape conditions. Both timber simulations were run for a 20-year period, and results are standardized to the current landscape. Because there is no timber value prior to the timber harvest simulations, the clearcut simulation was assigned a value of 100% for timber value.

system allows users to understand the "production curves" of ES and other socioeconomic indicators (Figs. 13 and 14) to identify the optimal level of a given management action.

## Modelling human decision-making behaviour

Modelling human behaviour is complex, and models are unlikely to capture all of the factors that real-world decisions will be based on. For example, mill agents in the timber model base their decisions on where to harvest based on the transportation costs, in order to maximize profits. However, decisions behind harvest siting are typically more complex, considering factors such as environmental performance (*Alberta-Pacific Forest Industries, 2007*) or integrating landscape planning with other resource companies (*Government of Alberta, 2015*).

In addition to external factors motivating human behaviour, there may also be complex feedbacks and interdependencies among multiple actors in a given scenario. For example, in our agricultural expansion analysis, we treated all cells as independent decision-makers; however, cell- or field-level decisions will be interdependent for a number of reasons. This includes multiple cells being owned by the same individual, and the opportunity for a crop field to benefit from "free-riding" on pollinator-friendly management implemented in neighbouring cells, with the cost of that management borne by another landowner.

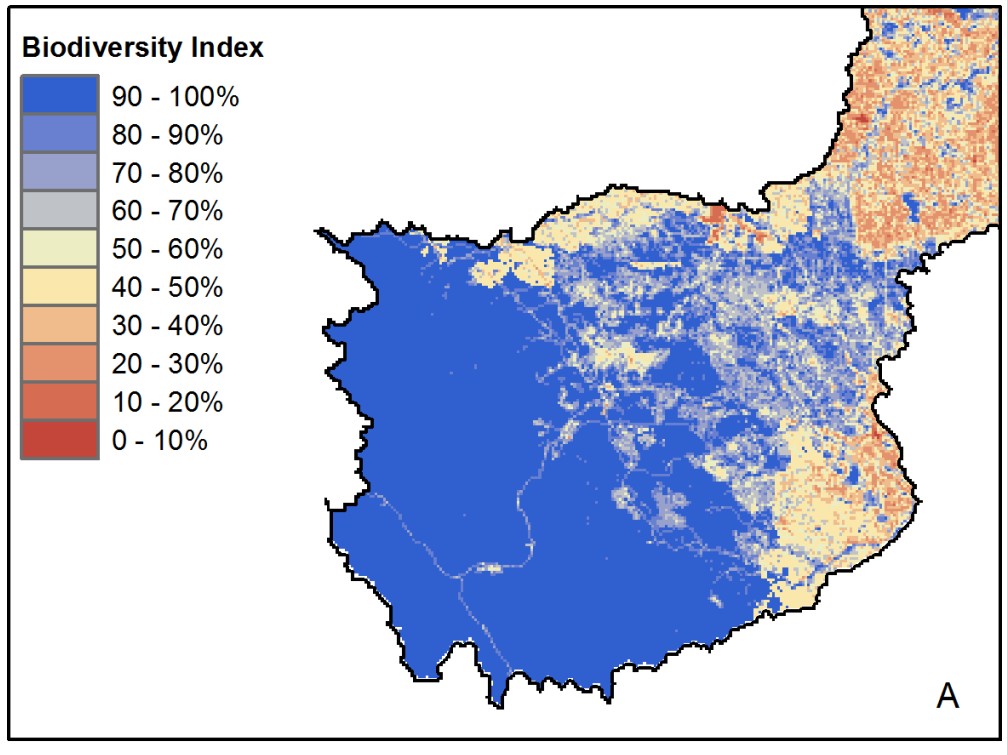

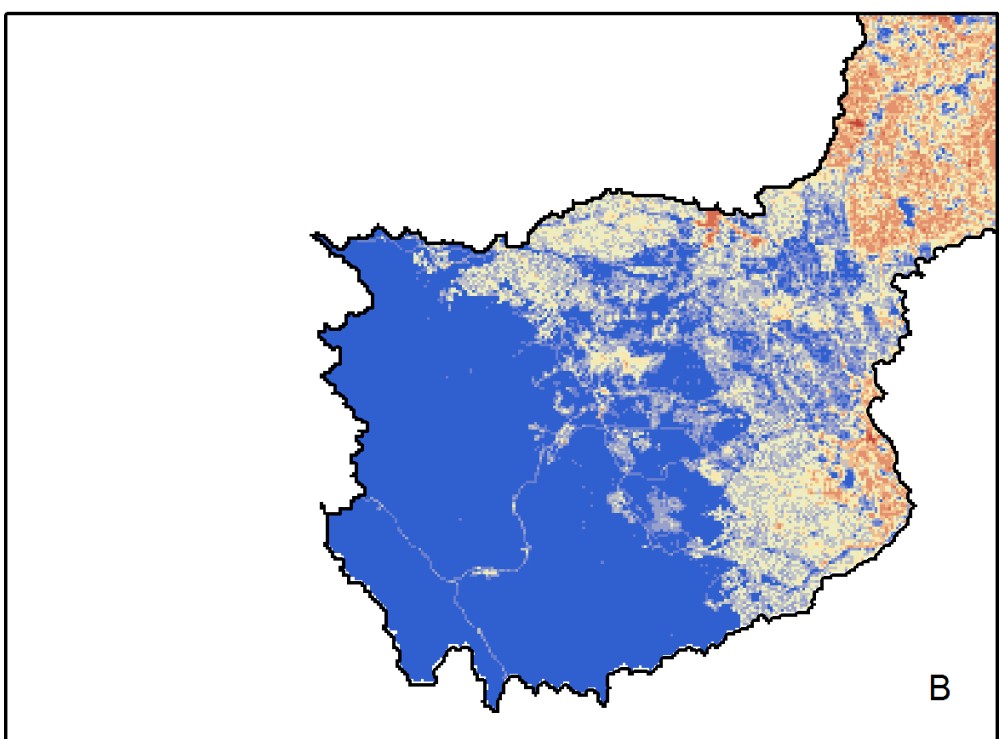

**Figure 11** **Modelled biodiversity index following 20-years of simulated timber harvest.** Map represents the predicted biodiversity responses to (A) standard clearcutting practices, and (B) variable retention timber harvest. Map is restricted to the active forestry area of the North Saskatchewan watershed.

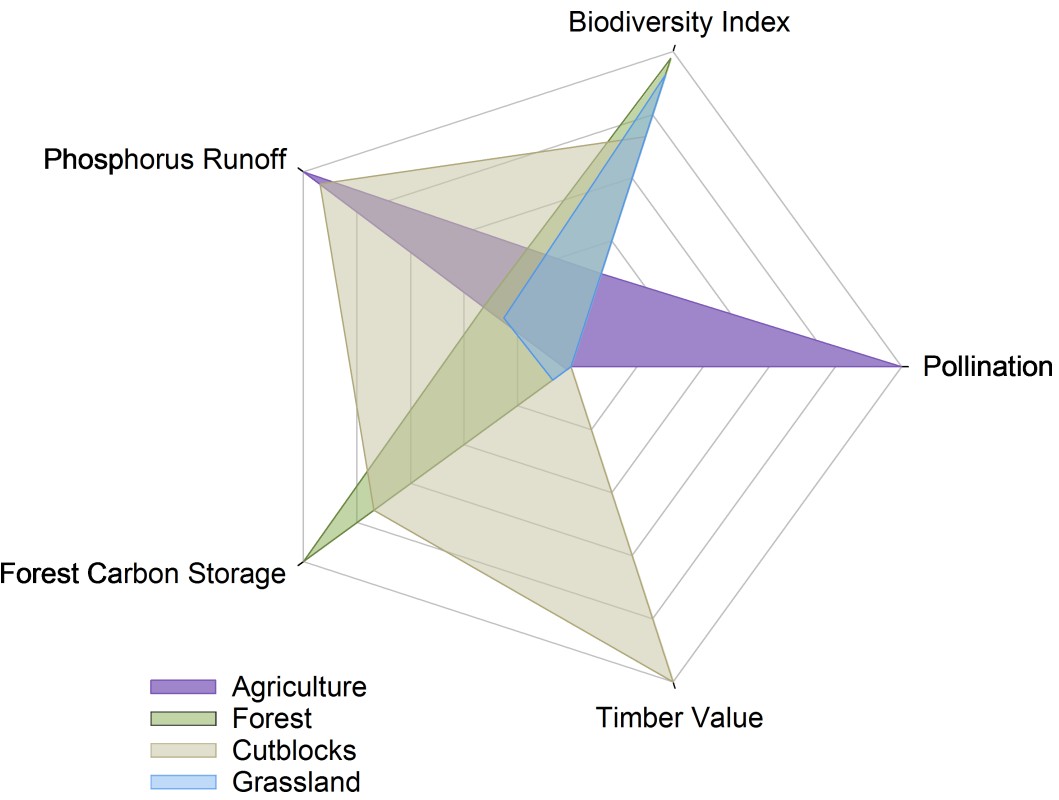

**Figure 12** **Suite of ecosystem services provided by four representative 100 km² landscapes.** All axes represent scores standardized from 0–100%. For pollination, timber, carbon, and phosphorus runoff, the highest value obtained from the four sample landscapes was artificially set as 100%. The biodiversity index was obtained directly from the model.

While we did not model these complex interdependencies in the current analysis, agent-based modelling platforms are an appropriate tool for simulating such a system (*Cong et al., 2014*). Our modelling package could be further developed to address them in the future, while also providing valuable information on how other socio-ecological indicators are affected as a result of agricultural land-use management decisions (e.g., Fig. 14).

## Applying ecosystem service models to land-use planning

While this modelling system can provide guidance on how to achieve a variety of ES-related goals, it requires complementary information from other sources to fully address most questions. For example, in our study region of central Alberta, blue–green algae blooms in large, recreational lakes are relatively common during summer months (*Pick, 2016*). The water model output layers can be analyzed solely within the watershed of a lake of interest to identify important areas of non-point source phosphorus pollution that are responsible for the algal blooms (*Trimbee & Prepas, 1987*). However, determining whether a given level of phosphorus reduction will be sufficient to eliminate algal blooms requires a detailed understanding of in-lake water chemistry (*Pick, 2016*), which a regional model such as ours cannot provide. Similarly, understanding the monetary benefits of eliminating algal

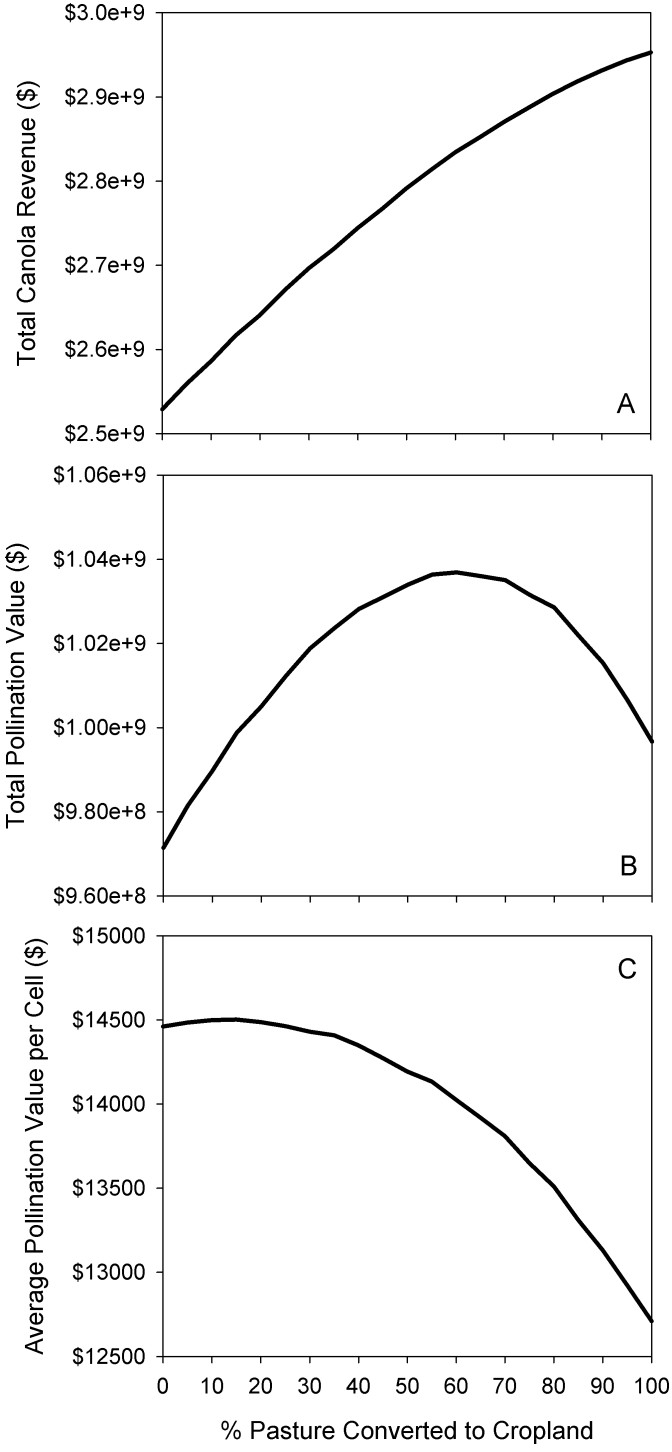

**Figure 13** **Effect of simulated agricultural expansion on pollinator value and total crop revenue.** Modelled response of canola revenue (A), total pollination value (B), and per-field pollination value (C) to increasing amounts of agricultural conversion in the North Saskatchewan watershed region of Alberta, Canada.

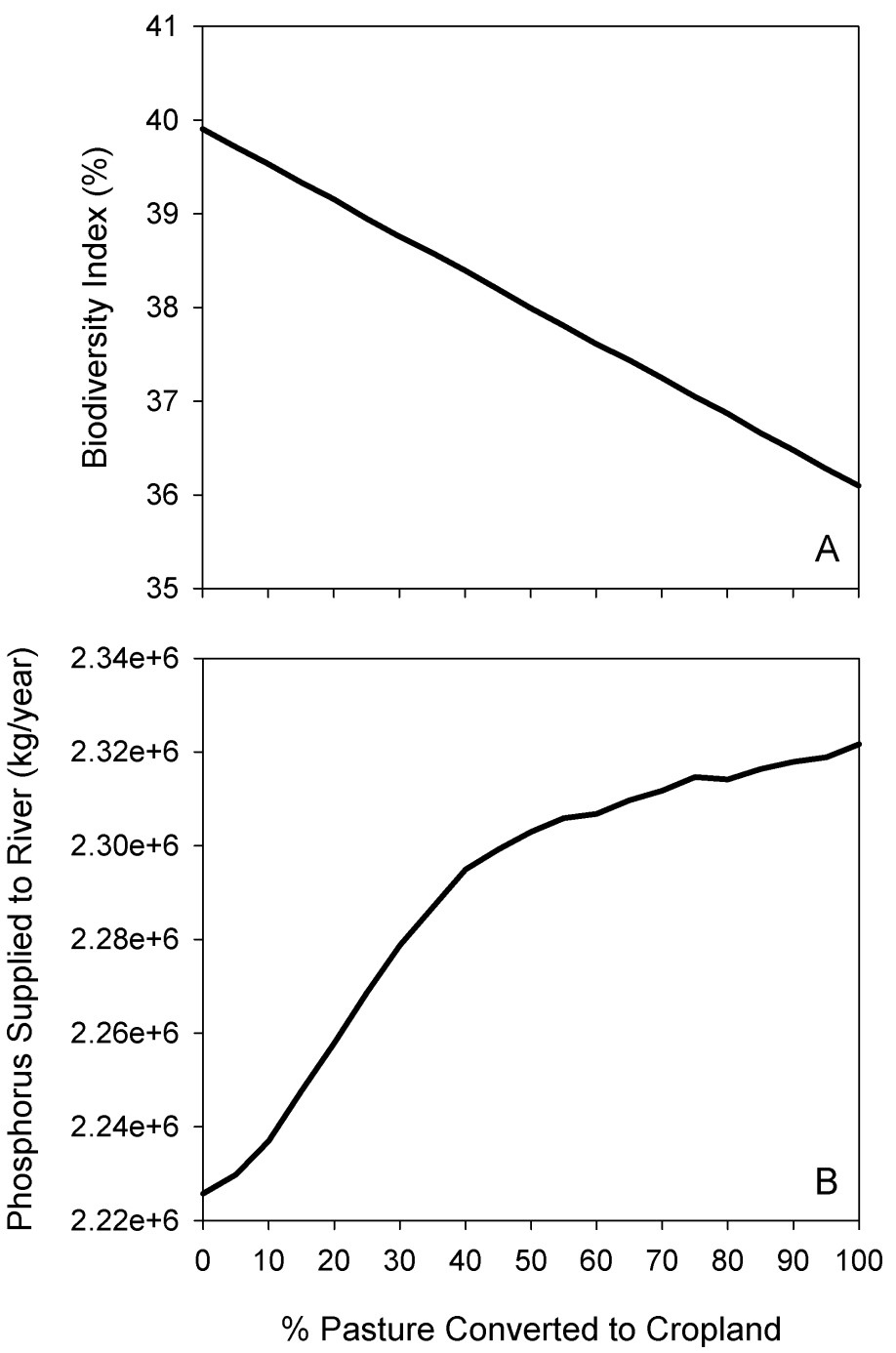

**Figure 14 Effect of simulated agricultural expansion on biodiversity and water quality.** Modelled response of biodiversity index (A) and phosphorus supply (B) to increasing amounts of agricultural conversion in the agriculture-dominated portion of the North Saskatchewan watershed region of Alberta, Canada.

blooms, such as increased property values and recreational tourism (*Keeler et al., 2012*), require detailed economic models; this information may be required for decision-makers to determine an acceptable budget for water quality improvement measures.

## CONCLUSIONS

We present an implementation of an ES modelling suite for a region in Alberta, Canada, but the majority of underlying modelling processes are generalizable to other jurisdictions if the appropriate data and GIS layers are available. Even the timber & carbon model, which draws on Alberta's unique FMA system, can largely be adapted to work in other jurisdictions where mills select specific cutblock locations from a larger forested area. In all cases of implementing this modelling suite in other jurisdictions, careful collection of data and parameter selection are essential.

Despite our focus on ES, including a separate model to evaluate biodiversity is necessary because ES are a poor proxy for biodiversity and vice versa. Despite the many linkages between biodiversity and ES, the relationship between biodiversity and a given ES is not always positive, or indeed, even present (*Cardinale et al., 2012*). In some cases, it may be most appropriate to consider the value of biodiversity as a component of other final ecosystem services, for example the additional tourism dollars from recreational bird watching in areas of high avian diversity. However, leaving aside these economic values, the intrinsic value of biodiversity is important to many stakeholders in its own right (*Soulé, 2013*), and cannot be addressed by the instrumental or economic valuation that is often the focal point of ES studies. Incorporating a separate assessment of biodiversity alongside the instrumental values of ecosystem services provides a safeguard against ignoring the intrinsic value of nature, while also providing information on indicators of interest to the widest possible range of stakeholders (*Tallis & Lubchenco, 2014*).

The ability to analyze landscapes and land-use management decisions using integrated, multi-ES models provides considerable insight into trade-offs between ES and other socioeconomic indicators. While the modelling suite we present is far from an exhaustive list of ecosystem services, it helps contribute to a fuller accounting of what is gained and lost through human land-use activities; future work to create additional assessment models, such as for grassland carbon storage, food production, and recreational values, would expand its utility. Likewise, although we present only two examples of land-use management decision-making, namely timber harvesting systems and agricultural expansion, the modelling suite can be further developed with additional, targeted sub-models to endogenously evaluate how other candidate land-use policies and actions can influence human decision-making and consequently ecosystem services. In particular, this ability to hard-wire human decisions into the model distinguishes this system from other existing ES modelling packages (Table 1; see Table 2 in *Bagstad et al., 2013b* for attributes of other modelling systems). By explicitly linking human decision-making and ES outcomes in a unified modelling platform, this modelling system allows stakeholders to evaluate the opportunities and trade-offs to make informed decisions for achieving land-use planning goals.

**Table 1  Attributes of the ecosystem services modelling system.** Table adapted from *Bagstad et al. (2013b)*.

| Criterion | Attributes |
| --- | --- |
| Quantifiable, approach to uncertainty | Quantitative, uncertainty though varying inputs (which can be automated in NetLogo) |
| Time requirements | Moderate to high, mostly for finding data and GIS pre-processing |
| Capacity for independent application | Yes |
| Level of development & documentation | Initial models complete and documented. Further model development is ongoing |
| Scalability | Regional, watershed, or landscape scale |
| Generalizability | High, with local data requirements |
| Nonmonetary & cultural perspectives | Biophysical values for all ES, and monetary values for some. Currently no cultural values. |
| Affordability, insights, integration with existing environmental assessment | Software and model code are free downloads. Most appropriate as a regional tool; relative rankings more reliable than exact numerical estimates. Map/GIS outputs of multiple ES values. Endogenously evaluate effects of landscape and management changes on ES. |

## ACKNOWLEDGEMENTS

We thank D Huggard for supplying biodiversity equations, and S Neufeld for providing water quality monitoring data.

### Funding

Funding was provided by Alberta Innovates—Bio Solutions (Grant # BIO-12-006), the Alberta Livestock and Meat Agency (Grant # 2012S007S), and Alberta Innovates Technology Futures. The funders had no role in study design, data collection and analysis, decision to publish, or preparation of the manuscript.

### Grant Disclosures

The following grant information was disclosed by the authors:
Alberta Innovates—Bio Solutions: BIO-12-006.
Alberta Livestock and Meat Agency: 2012S007S.
Alberta Innovates Technology Futures.

### Competing Interests

Scott Heckbert is an employee of Alberta Innovates, Jeffrey Wilson is an employee of Green Analytics, and Andrew Vandenbroeck is an employee of Silvacom Ltd. The remaining authors declare that they have no competing interests.

## Author Contributions

- Thomas J. Habib conceived and designed the experiments, performed the experiments, analyzed the data, contributed reagents/materials/analysis tools, wrote the paper, prepared figures and/or tables, reviewed drafts of the paper.
- Scott Heckbert, Jeffrey J. Wilson and Andrew J. K. Vandenbroeck conceived and designed the experiments, contributed reagents/materials/analysis tools, reviewed drafts of the paper.
- Jerome Cranston contributed reagents/materials/analysis tools, reviewed drafts of the paper.
- Daniel R. Farr conceived and designed the experiments, reviewed drafts of the paper.

## Data Availability

The code has been supplied as a Supplemental File.

## Supplemental Information

Supplemental information for this article can be found online at http://dx.doi.org/10.7717/peerj.2814#supplemental-information.

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
