# Peer review of "Impacts of land-use management on ecosystem services and biodiversity: an agent-based modelling approach"

_PeerJ, doi:10.7717/peerj.2814_

## Round 0.1 · original submission · Minor Revisions

The paper was positively reviewed by three reviewers . Please pay close attention to the comments of reviewer 3, which will help you improve the manuscript.

Reviewer 1 ·

Basic reporting

No Comments

Experimental design

No Comments

Validity of the findings

I found this to be a very strong paper in terms of its content and descriptions. It was very clear and direct. One area that I am curious in knowing more is how the simplified modeling approach utilized in this study compares to more complex approaches, in a direct sense (i.e., if you replaced the pollination component with a more complex version, how do the results change). I think this is addressed narratively throughout the paper, but a direct comparison would be interesting and (likely) strength the argument for using more simply applied and described model components as policy decision making tools.

·

Basic reporting

Basic reporting.
This manuscript presents an integrated modelling approach based on several ecosystem services to assess the impacts of land use change and guide land-use decisions.
The rationale for the study is clear and the MS is well-presented overall. I have found most sections clear and easy to follow.
Each model is described separately in the methods section. By doing this, the authors tend not to describe the connections and integrations between the different models. This is however later done in a section called “Model integration”. While reading each model separately, I made a comment about the linkages between models (e.g. carbon and biodiversity). I suggest that the authors clarify upfront that each model is first described individually and the linkages between models are explained later.

Experimental design

The temporal component of the modelling approach is sometimes not clear. Could you please clarify for each model how this was implemented (e.g. for water purification, was annual run-off calculated?)
Under the biodiversity index, I did not see how the expected abundance was calculated under reference conditions. Could you please elaborate?

Validity of the findings

Like most models of ES, the results are hard to validate. While the methods are sound, it would be good to have some measure of the accuracy of the models. I am aware that this is not possible for most models, but some suggestions on how to validate each model will be useful. If such models are to be used for land-use making decisions, an assessment of the validity of the results is critical.
The authors argue that the models presented here are simple and enable decision-making by practitioners. However, there is no assessment or discussion of the simplicity of these models or comparison against other modelling platforms. I suggest adding a section to discuss how simple these models are based on key criteria (e.g. no of parameters for each model, level of expertise and data required, modelling skills required) and a comparison with existing platforms. While I appreciate the integration of the different models into one platform, it seems to be that the models are already relatively complex and require specialized skills and knowledge.

Additional comments

An interesting modelling platform

·

Basic reporting

L 375: present results in same order they are listed in methods (or vice versa)

L 405-450: Forest timber & carbon vs. Water purification vs. Pollination vs Biodiversity sections. I'm confused about the organization of these sections. Why are timber and biodiversity reported in terms of different forest management, but carbon and water purification just reported in terms of the effects of logging (presumably clear-cutting?), and pollination just reported in terms of current value? What about the effect of variable retention logging for water purification and carbon sequestration? And I realize pollination is not responsive to logging rate but it seems weird to have these sections arranged so they're reporting such different things. Can you have a section on "current values" another section on "effects of forest management" and another on "land-use change scenarios"? Or at least have it parallel the three goals laid out in L 351-357, even though the rationale for those is not completely clear to me (see comment in experimental design section).
(I also dislike this structure for the discussion. I think the discussion should be more all encompassing, of the main takeaways, rather than partitioned into specific sections, especially these sections that seem arbitrary and counter to the integration that is the whole point of this exercise!)

L 457-466: This is a great point to make, but it is not what illustrates the need for a "broad, comprehensive modeling approach" -- or at least not what I think "broad and comprehensive" implies, which is the need for running multiple services together to understand trade-offs. Rather, this is a very good example of why modeling both supply and demand together is so important. Also, L 466-469 seems to come out of nowhere. This is another important point to make but unrelated to the rest of the paragraph and should go somewhere else.

Figs. 6, 8 and 12 are difficult to see any difference in the chosen color scale - can they be rescaled, or just use the color scheme from Fig. 13? Having so many different color schemes is actually kind of distracting - would be easier for the reader to pick one and stick with it throughout.

Tallis & Lubchenco should be Tallis et al. - there are about a hundred authors on that paper

Experimental design

L78-83: Definition of agent-based modeling is confusing to me; these examples sound like they could be any spatially explicit model (they could be describing InVEST, which is not an ABM). What makes ABM different than other ES models?

L 145: 800m seems like a rather coarse scale for pollination and water purification. I can understand that it makes sense administratively but can it represent biophysical processes?

L 231: what was the resolution of the DEM? (Presumably not 800 m, since there are 90 m globally available)

L304: clarify what comprises "uncultivated" land. Is this undeveloped (as in natural or semi natural) or just noncrop (meaning residential could be included)?

L 334-335: what was the fit of the regression model?

L 350-371: I don't understand the rationale for the services or scenarios chosen, if it doesn't make sense to include the full set of services in the scenario analysis. For that matter, why isn't timber value considered in a trade-off assessment with pollination and water quality with the different harvest rates? The application of these models seems somewhat haphazard.

L 529-534: I'm confused - I thought this was what the pollination model was capturing. How else do you see the diminishing returns of converting to canola, if it's not accounting for the loss of individual cells of habitat - at the margin?

Validity of the findings

L 378-384: to a certain extent these findings are fairly obvious -- of course timber value goes up with harvest level and pollination is only valuable (and timber isn't) when there are crops present. What I find more interesting and should be stressed more is the trade-offs-- that a large increase in timber value of the higher harvest level cost only small decreases in water quality, carbon storage and biodiversity. What are you basing the "large" or "small" on- is it relative to itself (% of increase or decrease in service)? These numbers would be helpful to list out. One can qualitatively see this in Fig 3, but there are no scales on the diagram. How much does each contour line represent? Also, the caption either needs to be much more detailed or the text does, because if a reader has never seen a spider diagram before they will have no idea what this means. I was even stumped at first because I was looking at the difference between each LU type and the regional average (the blue), rather than the difference between the LU types (the green and the brown). This needs to be walked through more carefully if it's going to really support the results

L 385-387: Is pasture really that good of pollinator habitat? Is that what the Morandin study showed? If so this needs to be stated in the methods because it came as a surprise to me that that was what was providing the pollination value. Rangelands properly managed certainly have the potential to support flowering plants but so do croplands (if properly managed!) and "pasture" evokes a more intensive system

L 390: these seem like tiny numbers for 40% conversion! only 9% increase in revenue for 40% more land in production?? Or is it 40% conversion of remaining habitat and there's just not that much left?just stating how much pasture area there is relative to canola area would be a big help.

We know for a lot of these benefits (water purification, pollination, biodiversity) the spatial pattern of habitat conversion matters a lot. Can you say something about the rationale for the random pattern of agricultural expansion and what it looked like in practice (based on configuration of pasture relative to cropland)?

L 405-409: Does this increase in sequestration with clear-cutting assume no loss in below ground or soil carbon from clear-cutting? And no long term emissions from harvested wood products (if a certain amount is thrown away/winds up in landfill)? State the assumptions more clearly. -- OK I see that you address this in the discussion, but I think it might go better in the methods so people wondering this when you report the results will have already seen it

L 422-423: Did you quantify retention in watersheds around major lakes as well? If not, just change the order of this to come before the identification of city drinking water as the only beneficiary modeled so that it doesn't leave it hanging so awkwardly

L 518-519: I don't really understand this claim. How is it different than other water quality models (like SWAT) - seems like those also "link land uses to water quality outcomes"? Also I appreciate the discussion on the many different benefits of water quality, but the only beneficiary in this study (I think?) was the city drinking water. If this model is to be taken up by government or other decision makers in the region, it would be helpful to point out some of the data sources or what information specifically you would need to be able to address these other water quality uses (fishing, property values, human health).

L 541-542: This is another unsubstantiated claim. How is this model "well-suited to addressing these questions" of interdependencies if, as you note in L 535-536, all cells are treated as independent decision makers?

L 574- 578: This seems like a really big deal! Isn't part of the point to compare clear-cutting to variable retention logging practices? If the model is unfairly penalizing variable retention logging to make it look worse than clear-cutting, then it probably shouldn't be used to compare the two! Am I misinterpreting? I would suggest some more justification here, or removing the comparison between the two logging practices for biodiversity in the main results.

Additional comments

This is a very thorough and thoughtful assessment of several ecosystem services across Alberta. Many interesting issues are addressed, including representation of the use or demand of ES as well as the supply and understanding trade-offs between different ES under different management regimes, but at times the dots don't seem well enough connected to support the conclusions (see specific comments in other sections). I have a few structural suggestions for the results and discussion (in the basic reporting section), but overall find it a very interesting and rich approach. One thing I would consider adding somewhere in the discussion or conclusion is the usability of the model, and what it would take for uptake by the decision-makers it is presumably intended to inform. Were they involved in the development at all? Is there a sense of whether this can answer the real questions they're facing? Addressing these points explicitly would take this from a very theoretically interesting paper to a more impactful one, linking science to policy action.

---

## Round 0.2 · accepted · Accept

Clearly you went beyond minor revisions and greatly improved the paper in response to reviewer comments. Your effort is greatly appreciated by PeerJ. I think your paper will be of wide interest.